# Dataset: Biodiversity of Ground Beetles (Coleoptera, Carabidae) of the Republic of Mordovia (Russia)

**Leonid V. Egorov** [1,2], **Viktor V. Aleksanov** [3], **Sergei K. Alekseev** [3], **Alexander B. Ruchin** [2,*], **Oleg N. Artaev** [4], **Mikhail N. Esin** [2], **Sergei V. Lukiyanov** [2,5], **Evgeniy A. Lobachev** [2,5] **and Gennadiy B. Semishin** [2]

1    Prisursky State Nature Reserve, 428034 Cheboksary, Russia; platyscelis@mail.ru
2    Joint Directorate of the Mordovia State Nature Reserve and National Park «Smolny», 430005 Saransk, Russia; esinmishka@gmail.com (M.N.E.); lukiyanovs@gmail.com (S.V.L.); lobachevea@ya.ru (E.A.L.); g.semishin@mail.ru (G.B.S.)
3    Parks Directorate of Kaluga Region, 248000 Kaluga, Russia; victor_alex@list.ru (V.V.A.); stenus@yandex.ru (S.K.A.)
4    Papanin Institute for Biology of Inland Waters, Russian Academy of Sciences, 152742 Borok, Russia; artaev@gmail.com
5    National Research Mordovia State University, 430005 Saransk, Russia
*    Correspondence: ruchin.alexander@gmail.com; Tel.: +7-83445-296-35

**Abstract:** (1) Background: Carabidae is one of the most diverse families of Coleoptera. Many species of Carabidae are sensitive to anthropogenic impacts and are indicators of their environmental state. Some species of large beetles are on the verge of extinction. The aim of this research is to describe the Carabidae fauna of the Republic of Mordovia (central part of European Russia); (2) Methods: The research was carried out in April-September 1979, 1987, 2000, 2001, 2005, 2007–2022. Collections were performed using a variety of methods (light trapping, soil traps, window traps, etc.). For each observation, the coordinates of the sampling location, abundance, and dates were recorded; (3) Results: The dataset contains data on 251 species of Carabidae from 12 subfamilies and 4576 occurrences. A total of 66,378 specimens of Carabidae were studied. Another 29 species are additionally known from other publications. Also, twenty-two species were excluded from the fauna of the region, as they were determined earlier by mistake (4). Conclusions: The biodiversity of Carabidae in the Republic of Mordovia included 280 species from 12 subfamilies. Four species (*Agonum scitulum*, *Lebia scapularis*, *Bembidion humerale*, and *Bembidion tenellum*) were identified for the first time in the Republic of Mordovia.

**Dataset:** https://doi.org/10.15468/5zvf4v. Accessed on (12 October 2023)

**Dataset License:** Creative Commons Attribution (CC-BY) 4.0 License

**Keywords:** dataset; coleoptera; carabidae; data paper; occurrences; Republic of Mordovia



## 1. Summary

The study of invertebrate biodiversity is still relevant for modern faunistic [1–3]. Changes in the structure of the habitat as a result of human activity are considered the greatest threats to biodiversity [4]. In many parts of the world, spatial patterns of habitat location and landscape structure have significantly changed as a result of ecosystem destruction and land-use intensification [5–7]. This has a significant impact on the biodiversity and structure of local communities [8]. One of the many causes of biodiversity loss is habitat modification, mainly because of the transformation of the natural landscape into agriculture [9]. The reduction and isolation of wild species can lead to the loss of biodiversity as a result of species extinction. A decrease in the size of habitat areas (fragmentation) and an increase in isolation between fragments changes the species richness and abundance

of species, thereby changing the structure of the community [7,10]. Changes in climatic conditions, including aridization, and related secondary causes, such as fires, droughts, and floods, influence the loss of diversity [11–13]. Toxic chemical pollution, urbanization, deforestation, and the introduction of invasive species have recently significantly impacted regional biodiversity [14–18].

Although Coleoptera have been studied better than many other invertebrate groups, their regional fauna in central European Russia has not yet been fully identified. Ground beetles (Carabidae) are one of the largest families of Coleoptera, represented by many species in almost all terrestrial biocenoses. The majority of species live in soil and ground substrates, and few representatives are found under the bark of trees and on herbaceous plants [19–21]. Depending on edaphic conditions, humidity, relief, microclimate, and vegetation cover, certain species compositions of ground beetles have been established [22–25]. Therefore, they can serve as excellent indicators of the ecological conditions of biocenoses and are widely used in monitoring studies [26–29], including in specially protected natural areas [30,31]. Nowadays, many datasets have been published on the beetles in European Russia, including the western [32,33], southern [34,35], and eastern [36] regions. The Republic of Mordovia, occupying an intermediate position between the western and eastern regions of European Russia, is of great interest in faunal studies. Datasets were published earlier on the beetles of the largest protected natural areas, such as the Mordovia State Nature Reserve and National Park "Smolny" [37,38]. However, the diversity of the carabid fauna in this region is far from being limited to them.

The purpose of this study was to describe the fauna in the form of modern data on the occurrence of Carabidae (Coleoptera) in the Republic of Mordovia [39].

## 2. Data Description

### 2.1. Data Set Name

Each observation includes basic information, such as location (latitude/longitude), date of observation, observer name, and identifier name. Coordinates were determined in the field using a GPS device or after surveys using Google Maps (Table 1). A total of 66,378 specimens were studied.

**Table 1.** Description of data in the dataset.

| Column Label | Column Description |
| --- | --- |
| eventID | An identifier for the set of information associated with an Event (occurs in one place at one time). |
| occurrenceID | An identifier for the Occurrence (as opposed to a particular digital record of the occurrence). |
| basisOfRecord | The specific nature of the data record: HumanObservation |
| scientificName | The full scientific name, including the genus name and the lowest level of taxonomic rank with the authority |
| kingdom | The full scientific name of the kingdom in which the taxon is classified |
| phylum | The full scientific name of the phylum or division in which the taxon is classified |
| class | The full scientific name of the class in which the taxon is classified |
| order | The full scientific name of the order in which the taxon is classified |
| taxonRank | The taxonomic rank of the most specific name in the scientificName. |
| decimalLatitude | The geographic latitude of location in decimal degree |
| decimalLongitude | The geographic longitude of the location in decimal degrees |
| geodeticDatum | The ellipsoid, geodetic datum, or spatial reference system (SRS) upon which the geographic coordinates are given in decimalLatitude and decimalLongitude as based. Here—WGS84. |
| coordinateUncertaintyInMeters | The horizontal distance (in meters) from the given decimalLatitude and decimalLongitude describing the smallest circle containing the whole of the Location |
| country | The name of the country in which the Location occurs. Here—Russia. |
| countryCode | The standard code for the country in which the Location occurs. Here—RU. |
| individualCount | The number of individuals represented present at the time of the Occurrence. |
| eventDate | The date when material from the trap was collected or the range of dates during which the trap collected material |
| year | The integer day of the month on which the Event occurred. |
| month | The ordinal month in which the Event occurred. |
| day | The integer day of the month on which the Event occurred |
| recordedBy | A person or group responsible for recording the original Occurrence. |
| identifiedBy | A list of names of people who assigned the Taxon to the subject |

## 2.2. Figures, Tables, and Schemes

The dataset presented data on 251 species of Carabidae from 12 subfamilies studied during our research (Table 2). In addition, Table 2 includes another 29 species of Carabidae (Table 2), which have been reported in other publications [29,37,40–43].

**Table 2.** Biodiversity of Carabidae species in the Republic of Mordovia.

| Subfamily, Species | Approximate Estimate of the Species Abundance |
|---|---|
| **Brachininae** | |
| *Brachinus crepitans* (Linnaeus, 1758) | single individual |
| *Brachinus nigricornis* Gebler, 1830 | single individual |
| **Carabinae** | |
| *Calosoma inquisitor* (Linnaeus, 1758) | common species |
| *Calosoma investigator* (Illiger, 1798) | rare species |
| *Calosoma maderae* (Fabricius, 1775) | rare species |
| *Calosoma sycophanta* (Linnaeus, 1758) | rare species |
| *Carabus arvensis baschkiricus* Breuning, 1932 | numerous species |
| *Carabus cancellatus* Illiger, 1798 | numerous species |
| *Carabus clathratus* Linnaeus, 1761 | rare species |
| *Carabus convexus* Fabricius, 1775 | numerous species |
| *Carabus coriaceus* Linnaeus, 1758 | common species |
| *Carabus estreicheri* Fischer von Waldheim, 1820 | single individual |
| *Carabus glabratus* Paykull, 1790 | numerous species |
| *Carabus granulatus* Linnaeus, 1758 | numerous species |
| *Carabus hortensis* Linnaeus, 1758 | numerous species |
| *Carabus nemoralis* O.F. Müller, 1764 | common species |
| *Carabus nitens* Linnaeus, 1758 | single individual |
| *Carabus schoenherri* Fischer von Waldheim, 1820 | single individual |
| *Carabus stscheglowi* Mannerheim, 1827 | single individual |
| *Carabus violaceus aurolimbatus* Dejean, 1830 | single individual |
| *Cychrus caraboides* (Linnaeus, 1758) | common species |
| **Cicindelinae** | |
| *Cicindela campestris* Linnaeus, 1758 | common species |
| *Cicindela hybrida* Linnaeus, 1758 | common species |
| *Cicindela maritima* Dejean, 1822 | single individual |
| *Cicindela sylvatica* Linnaeus, 1758 | common species |
| *Cicindela soluta* Dejean, 1822 | single individual |
| *Cylindera germanica* (Linnaeus, 1758) | common species |
| **Broscinae** | |
| *Broscus cephalotes* (Linnaeus, 1758) | common species |
| *Miscodera arctica* (Paykull, 1798) | single individual |
| **Elaphrinae** | |
| *Blethisa multipunctata* (Linnaeus, 1758) | single individual |
| *Elaphrus cupreus* Duftschmid, 1812 | common species |
| *Elaphrus riparius* (Linnaeus, 1758) | rare species |
| *Elaphrus uliginosus* Fabricius, 1792 | single individual |
| **Harpalinae** | |
| *Acupalpus elegans* (Dejean, 1829) | rare species |
| *Acupalpus exiguus* Dejean, 1829 | single individual |
| *Acupalpus flavicollis* (Sturm, 1825) | single individual |
| *Acupalpus meridianus* (Linnaeus, 1761) | common species |
| *Acupalpus parvulus* (Sturm, 1825) | single individual |
| *Agonum dolens* (C.R. Sahlberg, 1827) | single individual |
| *Agonum ericeti* (Panzer, 1809) | single individual |
| *Agonum hypocrita* (Apfelbeck, 1904) | single individual |
| *Agonum fuliginosum* (Panzer, 1809) | common species |
| *Agonum gracile* Sturm, 1824 | common species |
| *Agonum gracilipes* (Duftschmid, 1812) | common species |
| *Agonum impressum* (Panzer, 1796) | single individual |

**Table 2.** *Cont.*

| Subfamily, Species | Approximate Estimate of the Species Abundance |
|---|---|
| *Agonum lugens* (Duftschmid, 1812) | common species |
| *Agonum marginatum* (Linnaeus, 1758) | single individual |
| *Agonum micans* (Nicolai, 1822) | rare species |
| *Agonum muelleri* (Herbst, 1784) | single individual |
| *Agonum piceum* (Linnaeus, 1758) | single individual |
| * *Agonum scitulum* Dejean, 1828 | single individual |
| *Agonum sexpunctatum* (Linnaeus, 1758) | common species |
| *Agonum thoreyi* Dejean, 1828 | single individual |
| *Agonum versutum* Sturm, 1824 | rare species |
| *Agonum viduum* (Panzer, 1796) | rare species |
| *Agonum viridicupreum* (Goeze, 1777) | single individual |
| *Amara aenea* (De Geer, 1774) | numerous species |
| *Amara apricaria* (Paykull, 1790) | rare species |
| *Amara aulica* (Panzer, 1796) | numerous species |
| *Amara bifrons* (Gyllenhal, 1810) | numerous species |
| *Amara brunnea* (Gyllenhal, 1810) | numerous species |
| *Amara communis* (Panzer, 1797) | numerous species |
| *Amara consularis* (Duftschmid, 1812) | common species |
| *Amara convexior* Stephens, 1828 | single individual |
| *Amara convexiuscula* (Marsham, 1802) | single individual |
| *Amara crenata* Dejean, 1828 | single individual |
| *Amara curta* Dejean, 1828 | single individual |
| *Amara equestris* (Duftschmid, 1812) | common species |
| *Amara erratica* (Duftschmid, 1812) | single individual |
| *Amara eurynota* (Panzer, 1796) | common species |
| *Amara famelica* C.C.A. Zimmermann, 1832 | single individual |
| *Amara familiaris* (Duftschmid, 1812) | common species |
| *Amara fulva* (O.F. Müller, 1776) | rare species |
| *Amara gebleri* Dejean, 1831 | rare species |
| *Amara infima* (Duftschmid, 1812) | single individual |
| *Amara ingenua* (Duftschmid, 1812) | common species |
| *Amara littorea* C.G. Thomson, 1857 | rare species |
| *Amara lunicollis* Schiødte, 1837 | common species |
| *Amara majuscula* (Chaudoir, 1850) | common species |
| *Amara montivaga* Sturm, 1825 | rare species |
| *Amara municipalis* (Duftschmid, 1812) | single individual |
| *Amara nitida* Sturm, 1825 | common species |
| *Amara ovata* (Fabricius, 1792) | common species |
| *Amara plebeja* (Gyllenhal, 1810) | rare species |
| *Amara praetermissa* (C.R. Sahlberg, 1827) | rare species |
| *Amara quenseli silvicola* C.C.A. Zimmermann, 1832 | single individual |
| *Amara similata* (Gyllenhal, 1810) | common species |
| *Amara spreta* Dejean, 1831 | rare species |
| *Amara tibialis* (Paykull, 1798) | common species |
| *Anchomenus dorsalis* (Pontoppidan, 1763) | common species |
| *Anisodactylus binotatus* (Fabricius, 1787) | common species |
| *Anisodactylus nemorivagus* (Duftschmid, 1812) | common species |
| *Anisodactylus signatus* (Panzer, 1796) | common species |
| *Anthracus consputus* (Duftschmid, 1812) | single individual |
| *Badister bullatus* (Schrank, 1798) | common species |
| *Badister collaris* Motschulsky, 1844 | common species |
| *Badister dilatatus* Chaudoir, 1837 | rare species |
| *Badister lacertosus* Sturm, 1815 | common species |
| *Badister meridionalis* Puel, 1925 | single individual |
| *Badister peltatus* (Panzer, 1796) | rare species |

**Table 2.** *Cont.*

| Subfamily, Species | Approximate Estimate of the Species Abundance |
|---|---|
| *Badister sodalis* (Duftschmid, 1812) | common species |
| *Badister unipustulatus* Bonelli, 1813 | common species |
| *Bradycellus caucasicus* (Chaudoir, 1846) | rare species |
| *Calathus ambiguus* (Paykull, 1790) | rare species |
| *Calathus erratus* (C.R. Sahlberg, 1827) | numerous species |
| *Calathus fuscipes* (Goeze, 1777) | common species |
| *Calathus melanocephalus* (Linnaeus, 1758) | common species |
| *Calathus micropterus* (Duftschmid, 1812) | common species |
| *Callistus lunatus* (Fabricius, 1775) | common species |
| *Chlaenius nigricornis* (Fabricius, 1787) | rare species |
| *Chlaenius nitidulus* (Schrank, 1781) | single individual |
| *Chlaenius tristis* (Schaller, 1783) | rare species |
| *Chlaenius vestitus* (Paykull, 1790) | single individual |
| *Cymindis angularis* Gyllenhal, 1810 | common species |
| *Cymindis humeralis* (Geoffroy, 1785) | rare species |
| *Cymindis macularis* Fischer von Waldheim, 1824 | single individual |
| *Cymindis vaporariorum* (Linnaeus, 1758) | rare species |
| *Demetrias monostigma* Samouelle, 1819 | single individual |
| *Diachromus germanus* (Linnaeus, 1758) | single individual |
| *Dicheirotrichus rufithorax* (C.R. Sahlberg, 1827) | single individual |
| *Dolichus halensis* (Schaller, 1783) | common species |
| *Dromius agilis* (Fabricius, 1787) | single individual |
| *Dromius fenestratus* (Fabricius, 1794) | single individual |
| *Dromius quadraticollis* A. Morawitz, 1862 | single individual |
| *Dromius schneideri* Crotch, 1871 | single individual |
| *Harpalus affinis* (Schrank, 1781) | numerous species |
| *Harpalus anxius* (Duftschmid, 1812) | single individual |
| *Harpalus autumnalis* (Duftschmid, 1812) | single individual |
| *Harpalus calathoides* Motschulsky, 1844 | single individual |
| *Harpalus calceatus* (Duftschmid, 1812) | common species |
| *Harpalus distinguendus* (Duftschmid, 1812) | numerous species |
| *Harpalus flavescens* (Piller & Mitterpacher, 1783) | single individual |
| *Harpalus froelichii* Sturm, 1818 | rare species |
| *Harpalus griseus* (Panzer, 1796) | common species |
| *Harpalus hirtipes* (Panzer, 1796) | rare species |
| *Harpalus laevipes* Zetterstedt, 1828 | numerous species |
| *Harpalus latus* (Linnaeus, 1758) | numerous species |
| *Harpalus luteicornis* (Duftschmid, 1812) | common species |
| *Harpalus modestus* Dejean, 1829 | single individual |
| *Harpalus picipennis* (Duftschmid, 1812) | single individual |
| *Harpalus progrediens* Schauberger, 1922 | numerous species |
| *Harpalus pumilus* Sturm, 1818 | common species |
| *Harpalus rubripes* (Duftschmid, 1812) | numerous species |
| *Harpalus rufipes* (De Geer, 1774) | numerous species |
| *Harpalus signaticornis* (Duftschmid, 1812) | common species |
| *Harpalus smaragdinus* (Duftschmid, 1812) | common species |
| *Harpalus solitaris* Dejean, 1829 | single individual |
| *Harpalus subcylindricus* Dejean, 1829 | single individual |
| *Harpalus tardus* (Panzer, 1796) | numerous species |
| *Harpalus xanthopus winkleri* Schauberger, 1923 | common species |
| *Harpalus zabroides* Dejean, 1829 | common species |
| *Lebia chlorocephala* (J.J. Hoffmann, 1803) | rare species |
| *Lebia cruxminor* (Linnaeus, 1758) | common species |
| *Lebia cyanocephala* (Linnaeus, 1758) | single individual |

**Table 2.** *Cont.*

| Subfamily, Species | Approximate Estimate of the Species Abundance |
|---|---|
| *Lebia marginata* (Geoffroy, 1785) | single individual |
| * *Lebia scapularis* (Geoffroy, 1785) | single individual |
| *Licinus depressus* (Paykull, 1790) | common species |
| *Limodromus assimilis* (Paykull, 1790) | common species |
| *Limodromus krynickii* (Sperk, 1835) | common species |
| *Limodromus longiventris* Mannerheim, 1825 | single individual |
| *Masoreus wetterhallii* (Gyllenhal, 1813) | rare species |
| *Microlestes fissuralis* Reitter, 1901 | single individual |
| *Microlestes maurus* (Sturm, 1827) | common species |
| *Microlestes minutulus* (Goeze, 1777) | common species |
| *Odacantha melanura* (Linnaeus, 1767) | single individual |
| *Olisthopus rotundatus* (Paykull, 1790) | single individual |
| *Oodes gracilis* A. Villa & G.B. Villa, 1833 | single individual |
| *Oodes helopioides* (Fabricius, 1792) | common species |
| *Ophonus azureus* (Fabricius, 1775) | numerous species |
| *Ophonus laticollis* Mannerheim, 1825 | rare species |
| *Ophonus puncticollis* (Paykull, 1798) | common species |
| *Ophonus rufibarbis* (Fabricius, 1792) | common species |
| Ophonus rupicola (Sturm, 1818) | single individual |
| *Ophonus stictus* Stephens, 1828 | common species |
| *Oxypselaphus obscurus* (Herbst, 1784) | common species |
| *Panagaeus bipustulatus* (Fabricius, 1775) | common species |
| *Panagaeus cruxmajor* (Linnaeus, 1758) | single individual |
| *Paradromius linearis* (G.-A. Olivier, 1795) | common species |
| *Philorhizus sigma* (P. Rossi, 1790) | single individual |
| *Platynus livens* (Gyllenhal, 1810) | single individual |
| *Platynus mannerheimii* (Dejean, 1828) | single individual |
| *Poecilus crenuliger* Chaudoir, 1876 | single individual |
| *Poecilus cupreus* (Linnaeus, 1758) | numerous species |
| *Poecilus koyi* (Germar, 1823) | rare species |
| *Poecilus lepidus* (Leske, 1785) | numerous species |
| *Poecilus punctulatus* (Schaller, 1783) | rare species |
| *Poecilus versicolor* (Sturm, 1824) | numerous species |
| *Polystichus connexus* (Geoffroy, 1785) | single individual |
| *Pterostichus aethiops* (Panzer, 1796) | single individual |
| *Pterostichus anthracinus* (Illiger, 1798) | common species |
| *Pterostichus aterrimus* (Herbst, 1784) | single individual |
| *Pterostichus diligens* (Sturm, 1824) | rare species |
| *Pterostichus gracilis* (Dejean, 1828) | rare species |
| *Pterostichus macer* (Marsham, 1802) | common species |
| *Pterostichus mannerheimii* (Dejean, 1831) | single individual |
| *Pterostichus melanarius* (Illiger, 1798) | numerous species |
| *Pterostichus minor* (Gyllenhal, 1827) | common species |
| *Pterostichus niger* (Schaller, 1783) | numerous species |
| *Pterostichus nigrita* (Paykull, 1790) | numerous species |
| *Pterostichus oblongopunctatus* (Fabricius, 1787) | numerous species |
| *Pterostichus ovoideus* (Sturm, 1824) | single individual |
| *Pterostichus quadrifoveolatus* Letzner, 1852 | common species |
| *Pterostichus rhaeticus* Heer, 1837 | common species |
| *Pterostichus strenuus* (Panzer, 1796) | numerous species |
| *Pterostichus uralensis* (Motschulsky, 1850) | single individual |
| *Pterostichus vernalis* (Panzer, 1796) | common species |

**Table 2.** *Cont.*

| Subfamily, Species | Approximate Estimate of the Species Abundance |
|---|---|
| *Sericoda quadripunctata* (De Geer, 1774) | rare species |
| *Stenolophus mixtus* (Herbst, 1784) | common species |
| *Stenolophus teutonus* (Schrank, 1781) | rare species |
| *Stomis pumicatus* (Panzer, 1796) | common species |
| *Syntomus foveatus* (Geoffroy, 1785) | single individual |
| *Syntomus truncatellus* (Linnaeus, 1761) | common species |
| *Synuchus vivalis* (Illiger, 1798) | common species |
| **Loricerinae** | |
| *Loricera pilicornis* (Fabricius, 1775) | common species |
| **Nebriinae** | |
| *Leistus ferrugineus* (Linnaeus, 1758) | common species |
| *Leistus terminatus* (Panzer, 1793) | common species |
| *Nebria livida* (Linnaeus, 1758) | single individual |
| *Notiophilus aquaticus* (Linnaeus, 1758) | common species |
| *Nothiophilus aestuans* Dejean, 1826 | single individual |
| *Notiophilus biguttatus* (Fabricius, 1779) | single individual |
| *Notiophilus germinyi* Fauvel, 1863 | common species |
| *Notiophilus palustris* (Duftschmid, 1812) | numerous species |
| **Omophroninae** | |
| *Omophron limbatum* (Fabricius, 1777) | common species |
| **Patrobinae** | |
| *Patrobus assimilis* Chaudoir, 1844 | rare species |
| *Patrobus atrorufus* (Strøm, 1768) | common species |
| *Patrobus septentrionis* Dejean, 1828 | single individual |
| **Scaritinae** | |
| *Clivina fossor* (Linnaeus, 1758) | common species |
| *Dyschirius aeneus* (Dejean, 1825) | single individual |
| *Dyschirius angustatus* (Ahrens, 1830) | single individual |
| *Dyschirius globosus* (Herbst, 1784) | single individual |
| *Dyschirius nitidus* (Dejean, 1825) | single individual |
| *Dyschirius politus* (Dejean, 1825) | single individual |
| *Dyschirius thoracicus* (P. Rossi, 1790) | single individual |
| *Dyschirius tristis* Stephens, 1827 | single individual |
| *Dyschiriodes neresheimeri* (Wagner, 1915) | single individual |
| **Trechinae** | |
| *Asaphidion flavipes* (Linnaeus, 1761) | common species |
| *Asaphidion pallipes* (Duftschmid, 1812) | single individual |
| *Bembidion argenteolum* Ahrens, 1812 | single individual |
| *Bembidion articulatum* (Panzer, 1796) | common species |
| *Bembidion assimile* Gyllenhal, 1810 | single individual |
| *Bembidion azurescens* (Dalla Torre, 1877) | single individual |
| *Bembidion biguttatum* (Fabricius, 1779) | common species |
| *Bembidion bruxellense* Wesmael, 1835 | single individual |
| *Bembidion bualei polonicum* J. Müller, 1930 | single individual |
| *Bembidion decorum* (Panzer, 1799) | single individual |
| *Bembidion dentellum* (Thunberg, 1787) | rare species |
| *Bembidion doris* (Panzer, 1796) | rare species |
| *Bembidion femoratum* Sturm, 1825 | single individual |
| *Bembidion fumigatum* (Duftschmid, 1812) | single individual |
| *Bembidion gilvipes* Sturm, 1825 | single individual |
| *Bembidion guttula* (Fabricius, 1792) | rare species |
| * *Bembidion humerale* Sturm, 1825 | single individual |
| *Bembidion lampros* (Herbst, 1784) | common species |
| *Bembidion litorale* (G.-A. Olivier, 1790) | rare species |
| *Bembidion lunatum* (Duftschmid, 1812) | single individual |

**Table 2.** *Cont.*

| Subfamily, Species | Approximate Estimate of the Species Abundance |
|---|---|
| *Bembidion mannerheimii* C.R. Sahlberg, 1827 | rare species |
| *Bembidion minimum* (Fabricius, 1792) | rare species |
| *Bembidion obliquum* Sturm, 1825 | single individual |
| *Bembidion octomaculatum* (Goeze, 1777) | single individual |
| *Bembidion properans* (Stephens, 1828) | common species |
| *Bembidion punctulatum* Drapiez, 1820 | single individual |
| *Bembidion pygmaeum* (Fabricius, 1792) | single individual |
| *Bembidion quadrimaculatum* (Linnaeus, 1761) | common species |
| *Bembidion ruficolle* (Panzer, 1796) | single individual |
| *Bembidion schueppelii* Dejean, 1831 | rare species |
| *Bembidion semipunctatum* (Donovan, 1806) | rare species |
| *Bembidion striatum* (Fabricius, 1792) | single individual |
| * *Bembidion tenellum* Erichson, 1837 | single individual |
| *Bembidion tetracolum* Say, 1823 | single individual |
| *Bembidion varium* (G.-A. Olivier, 1795) | rare species |
| *Bembidion velox* (Linnaeus, 1761) | single individual |
| *Blemus discus* (Fabricius, 1792) | single individual |
| *Porotachys bisulcatus* (Nicolai, 1822) | single individual |
| *Tachys micros* (Fischer von Waldheim, 1828) | single individual |
| *Tachyta nana* (Gyllenhal, 1810) | single individual |
| *Trechoblemus micros* (Herbst, 1784) | single individual |
| *Trechus quadristriatus* (Schrank, 1781) | common species |
| *Trechus rivularis* (Gyllenhal, 1810) | single individual |
| *Trechus rubens* (Fabricius, 1792) | single individual |
| *Trechus secalis* (Paykull, 1790) | common species |

*—new species for the Republic of Mordovia; underlining highlights the names of species known from references [29,37,40–43] and are not included in the dataset.

Twenty-two species were excluded from the fauna records of the Republic of Mordovia. These include *Amara sabulosa* (Audinet-Serville, 1821) and *Agonum duftschmidi* J. Schmidt, 1994; *Bembidion andreae* (Fabricius, 1787); *Bembidion cruciatum* Dejean, 1831; *Calomera littoralis conjunctaepustulata* (Dokhtouroff, 1887); *Clivina collaris* (Herbst, 1784); *Corsyra fusula* (Fischer von Waldheim, 1820); *Bembidion foraminosum* (Sturm, 1825); *Harpalus amplicollis* Ménétriés, 1848; *Harpalus atratus* Latreille, 1804; *Harpalus dispar* Dejean, 1829; *Harpalus flavicornis* Dejean, 1829; *Harpalus politus* Dejean, 1829; *Harpalus pygmaeus* Dejean, 1829; *Harpalus saxicola* Dejean, 1829; *Ophonus cordatus* (Duftschmid, 1812); *Ophonus puncticeps* Stephens, 1828; *Paradromius longiceps* Dejean, 1826; *Philorhizus notatus* (Stephens, 1827); *Poecilus laevicollis* Chaudoir, 1842; *Poecilus puncticollis* Dejean, 1828; *Dicheirotrichus ustulatus* Dejean, 1829). Previously, they have been indicated in the publications of other authors [40–42,44–49]. These species have not been detected in our collections so far. Most of these species are also not recorded in neighbouring regions, which probably indicates, the erroneous identification of these species in the region. Thus, the total fauna of Carabidae in the Republic of Mordovia includes 280 species. Such species as *Agonum scitulum* Dejean, 1828, *Lebia scapularis* (Geoffroy, 1785), *Bembidion humerale* Sturm, 1825, and *Bembidion tenellum* Erichson, 1837 are new to the region and are included in this list.

Thus, the identified beetle fauna of the Republic of Mordovia includes 280 species, which is 14% of the known beetle fauna of Russia [50] and is close to the number of species in neighbouring regions located at the same latitude: the Republic of Tatarstan (303 species according to [51]), the Chuvash Republic (more than 270 species are data from the first author), and the Ryazan region (277 species according to [52]). High species richness, a large number of studied localities, a variety of collection methods, and the duration of the study make it possible to analyse the beetle fauna of Mordovia.

Regional fauna are often analysed using core and satellite hypotheses based on the analysis of species frequency distributions [53]. This approach has been successfully

applied to beetles [54,55]. For the obtained dataset, 15 species of beetles found in 50 or more localities were classified as core species in the Republic of Mordovia (Figure 1).

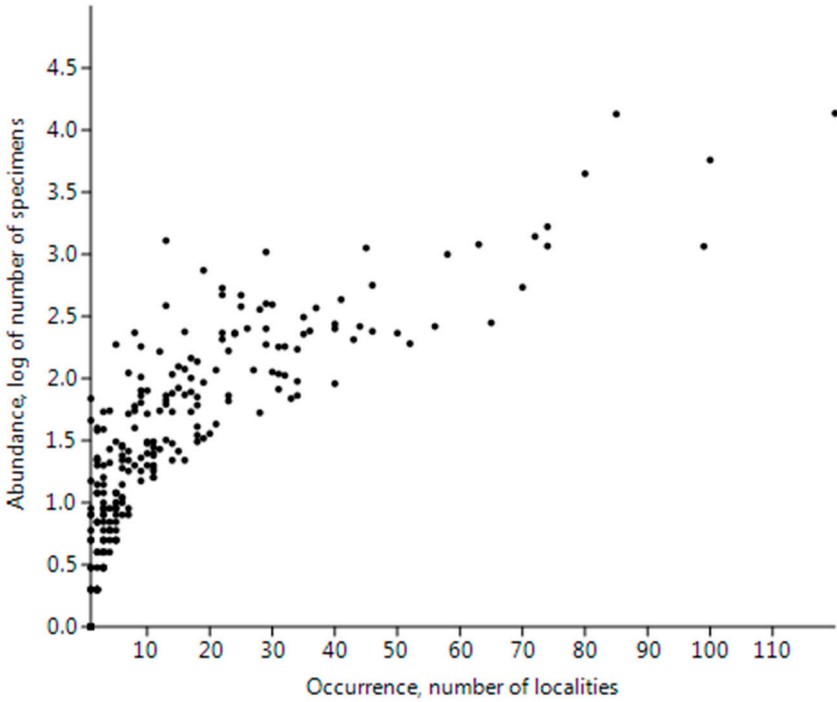

**Figure 1.** Occurrence—abundance distribution of ground beetle species in the Republic of Mordovia.

*Harpalus rufipes* had the highest occurrence level and was found at 120 localities (Figure 2). This accounted for almost 5% of all counted beetles. The high occurrence level and abundance of this species are probably due to both its ecological plasticity and good migratory ability [56–58], as well as its catchability using different collection methods. It is a mixophytophage capable of consuming both animal and plant food [59]. It has a polyvariant life cycle, hibernating at both larval and adult stages [60]. Due to this, *Harpalus rufipes* is a mass species in fields and gardens, but also occurs in floodplains [61] and forests, especially in anthropogenically disturbed forests [62].

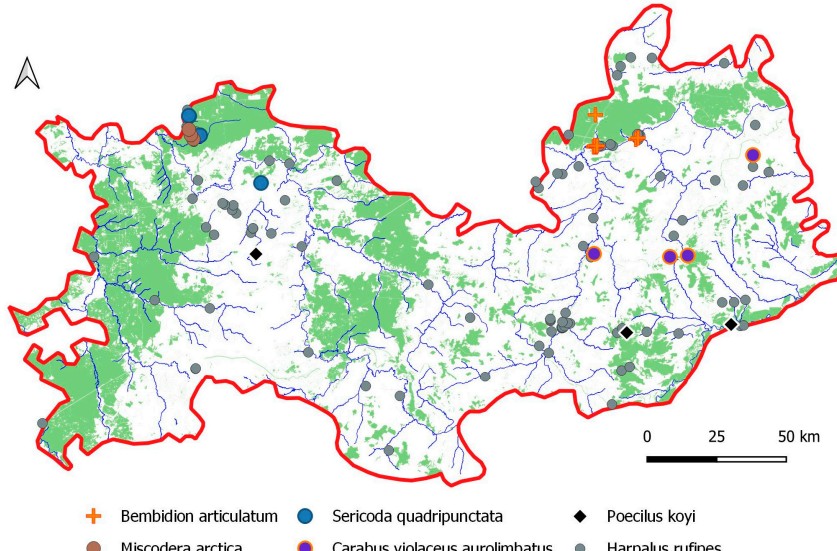

**Figure 2.** Distribution of the most frequent species, *Harpalus rufipes*, and some rare species in the Republic of Mordovia.

The second most common species were *Pterostichus melanarius* and *Pterostichus niger*. Both species are considered forest generalists and zoophages capable of facultatively consuming plant food [56]. They are zoophagous litter and soil-dwelling stratobionts [63]. The former species are well-adapted to life in the field because of their good burrowing ability, whereas the latter are able to travel long distances and occur in a variety of habitats via foot migration [64,65]. Both species have a polyvariant life cycle, which ensures that their populations are age-diverse and stable [66,67].

*Poecilus versicolor* belongs to the same group of life forms (zoophagous litter and soil-dwelling stratobionts), and is considered a characteristic of meadows. It is a mass species in some biotopes. *Poecilus cupreus*, which is close to its morphological and biological features, has a similar occurrence but does not reach such a high abundance. Additionally, among this group of life forms, *Pterostichus oblongopunctatus*, typical of forests, is at the core of the fauna. Among the smaller zoophagous litter-dwelling stratobionts, *Pterostichus strenuus* is the most abundant. Among the walking zoophagous epigeobionts, *Carabus cancellatus* and *Carabus granulatus* are at the core of the fauna. The first species in the zone of mixed and broad-leaved forests in the European part of Russia is usually considered an inhabitant of open biotopes [62]. However, it generally inhabits well-warmed forests [68].

Such Harpalus-like mixophytophage geohortobionts, such as *Harpalus rubripes*, *Harpalus latus*, *Amara aenea*, *Harpalus affinis*, *Amara communis*, and *Harpalus tardus*, are other species of the core fauna and inhabitants of open habitats. They are able to rapidly colonise suitable habitats by flight and consume both animal and plant food [63].

Thus, a peculiarity of the Republic of Mordovia is the predominance of open biotope species, including myxophytophagous species, in the core fauna, which distinguishes it from the western regions of the midlands [32,33]. In their core fauna, zoophagous and forest species occupy the main place. All representatives of the core are spring breeders *sensu latu*, or have polyvariant lifecycles.

Species with not very high occurrence but locally abundant include *Carabus arvensis baschkiricus*, *Limodromus assimilis*, and *Carabus nemoralis*. The first beetle is considered a xero-thermophilic species inhabiting well-warmed coniferous forests, clearings, heathlands [68]. *L. assimilis* is a forest species whose distribution is determined by soil moisture and forest litter [69]. The latter species in the European part of Russia is an inhabitant of anthro-pogenically disturbed forests and continues to disperse eastward [62,68]. In the Republic of Mordovia, it is characteristic of the forest habitats of Saransk and its surroundings.

The group of single individuals included a quarter of the entire species list in the dataset (62 out of 251). The reasons for their rarity vary. Some species, due to their biological peculiarities, are rarely caught in traps, especially in soil traps. These are, for example, *Trechoblemus micros* and *Dromius* spp. Others are apparently rare in this region. Among them, species with single occurrences but relatively high abundance, such as *Carabus stscheglowi*, and *Bembidion striatum*, require special attention from the point of view of protection.

This group of rare species is of great interest for analysis. Most of these species are found in different parts of the region with suitable habitats. However, some species, according to the data obtained, have a geographically limited distribution in the region (Figure 2).

Thus, *Miscodera arctica* is found only in the northwestern part of the Republic (Mordovia State Nature Reserve). In the European part of Russia, it is a stenotopic species inhabiting dry lichen pine forests [70]. *Serricoda quadripunctata* is found in the same part of the republic, but with a greater move towards the centre. *Bembidion articulatum* is found in the northeastern part (National Park "Smolny"). *Bembidion litorale* was found in both territories. These hygrophilous and mesohygrophilous species are probably limited by their confinement to certain biotopes and their limited migration abilities. At the same time, *Harpalus froelichi*, which is generally considered a thermophilous psammophilous species [71], was found only in the northeastern part of the Republic. *Carabus violaceus aurolimbatus*, a rare, predominantly forest-steppe taxon that gravitates to open landscapes, was recorded exclusively in the eastern part of the Republic [68]. Finally, *Poecilus koyi*, for

which very little data have been published, is located in the southern half of the Republic from northwest to southeast.

The data obtained can be used to clarify the configuration of the range of beetle species.

## 3. Methods

### 3.1. Study Area

The Republic of Mordovia is located at the junction of the Volga Upland and the Oka-Don Lowlands (Figure 3). The Volga Upland occupies the eastern part of the region and is hilly. The flat surfaces of the watershed massifs had absolute heights ranging from 280 to 320 m. Steep slopes are widespread, where the active demolition of weathering products takes place. The active development of erosion processes has resulted in a significantly dense gully beam network. The Oka-Don lowland is located in the western part of the region, constituting a lower and less hilly plain. The maximum absolute mark rarely exceeded 180 m. The lowlands have wide watershed spaces of up to 10 km and gentle slopes and are poorly dissected by ravines and gullies.

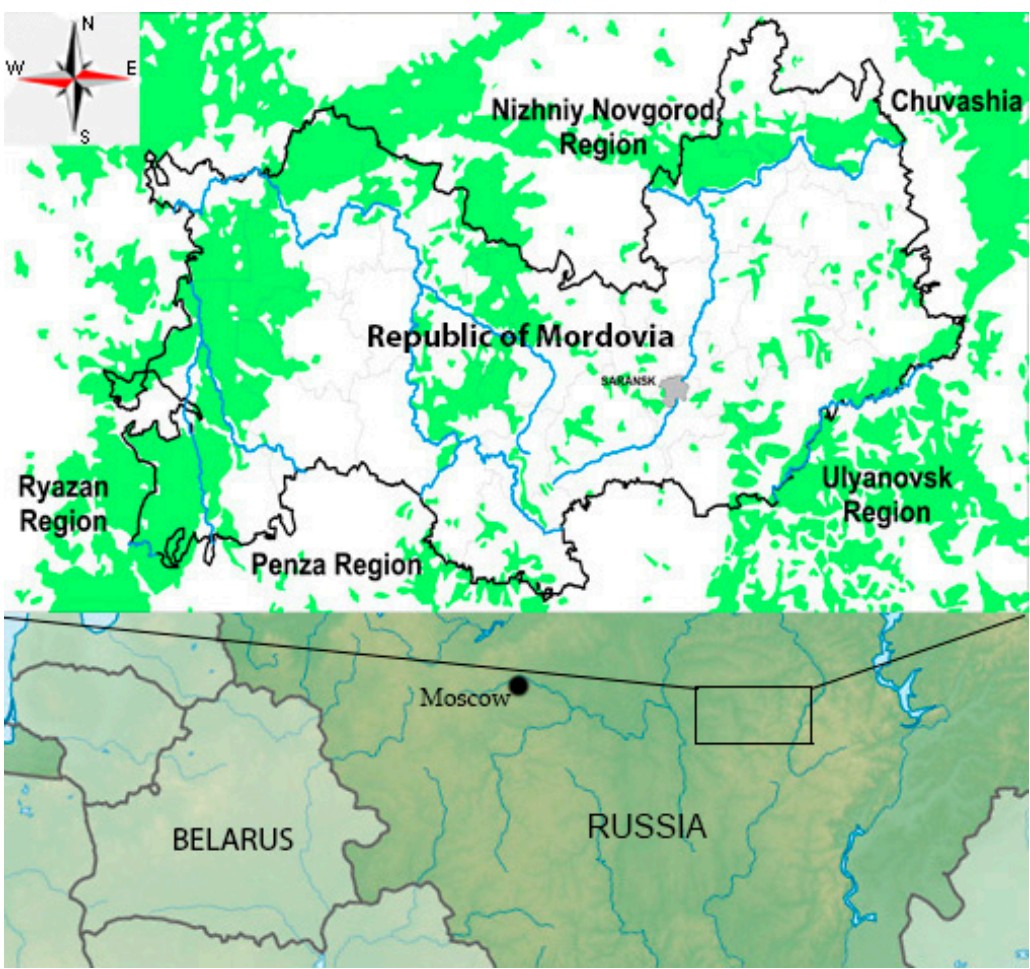

**Figure 3.** Study area for obtaining information from the dataset.

The average temperature of the coldest month (January) varied from −11.5 °C to −12.3 °C, and that of the warmest month (July) varied from 18.9 °C to 19.8 °C. Thus, the annual amplitude was 32.1 °C. The average annual air temperature varies from 3.5 °C to 4.0 °C. Three types of air masses participated in the formation of the main features of the climate: Arctic, temperate, and tropical, with a predominance of the second type. Air masses are represented by two varieties: continental and marine. Marine contains a large amount of moisture, and in the cold period, it often causes the formation of thaws, and in

the summer, cool weather. The average annual precipitation on the territory of Mordovia is 480 mm. During the long-term observations, periods of higher and lower moisture levels were observed. The deviation between the minimum and maximum values was 180 mm. During the year, precipitation prevailed during the warm period. From April to October, they fell to 80% of the annual norm. The average precipitation in July is approximately 65 mm, and the minimum monthly precipitation is 15–30 mm in February [72].

### 3.2. Design of Research, Identification and Taxonomic Position of Samples

We used traditional collection methods. We actively used a manual collection of samples using nets, pitfall traps, light fishing, window traps, pan traps, and partial beer traps [73,74]. Pitfall traps were most actively used. These traps were set during April-September 1979, 1987, 2000, 2001, 2005, 2007–2022 years. One trap was a 0.5-L plastic cup containing 200 mL of 4% formalin solution. In different biotopes, we installed from 10 to 20 such traps. The distance between the traps was 1.5–2.0 m. All samples were studied by S.K. Alekseev and L.V. Egorov. Identification was performed according to the methods described by Müller-Motzfeld [75] and Isaev [76]. We followed the proposed nomenclature in the works of Kryzhanovskii et al. [77], Lobl, and Lobl [78].

To estimate the abundance of each species listed in Table 2, the following definitions were used. "Single individual" means that single specimens of a species were found in no more than two localities in a region. "Rare species" refers to species with an abundance of up to 50 specimens occurring in 3–9 localities. "Common species" are species with an abundance of up to 100, found in more than 10 localities. "Numerous species" are Carabidae, with a total abundance of more than 100 specimens occurring in at least 20 percent of studied localities.

**Author Contributions:** Conceptualization, L.V.E.; methodology, A.B.R., V.V.A. and G.B.S.; software, O.N.A.; validation, A.B.R.; formal analysis, L.V.E., A.B.R. and V.V.A.; investigation, A.B.R., M.N.E., S.V.L., E.A.L. and G.B.S.; resources, A.B.R., S.K.A., M.N.E., S.V.L., E.A.L. and G.B.S.; data curation, S.K.A., M.N.E. and O.N.A.; writing—original draft preparation, A.B.R. and V.V.A.; writing—review and editing, L.V.E. and A.B.R.; visualization, A.B.R.; supervision, L.V.E. and V.V.A.; project administration, A.B.R.; funding acquisition, A.B.R. All authors have read and agreed to the published version of the manuscript.

**Funding:** This research was funded by Russian Science Foundation, grant number 22-14-00026.

**Institutional Review Board Statement:** Not applicable.

**Informed Consent Statement:** Not applicable.

**Data Availability Statement:** Data is contained within the article.

**Acknowledgments:** The authors are grateful for scientific advice and assistance in species identification by B.M. Kataev and I.I. Kabak (St. Petersburg, Russia).

**Conflicts of Interest:** The authors declare no conflict of interest.

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
