# Peer review of "Dataset: Biodiversity of Ground Beetles (Coleoptera, Carabidae) of the Republic of Mordovia (Russia)"

_data, 2023_

Round 1

Reviewer 1 Report

The article is a great contribution to science, I have some suggestions listed above (my suggestions are in italics):

P. 1 L. 15 - The aim of this research...

P. 1 L. 19 - coordinates of the sampling locations, abundance, and dates...

P. 1 L. 20 - ... subfamilies, and 4576 occurrences.

P. 9 L. 80 - ... and is not included in...

For the keywords I recommend the authors to choose some that are not in the title.

Fig. 2 and 3 could be joined in one and Fig. 4 could be better prepared, it is not in a level to be published in a scientific journal.

Following the Data journal guidelines, I did not find in the manuscript where the dataset is released, there is no supplementary material or link to access it. Also, one of the requisites is a DOI number, not added in the manuscript.

Another missing information was who will be responsible for updating the dataset.

The manuscript is well written, I just made some suggestions to improve the meaning.

Author Response

Dear reviewer. We are grateful for the critical analysis of our manuscript and your recommendations. We tried to make all the corrections. The DOI number is added on the first page after the annotation (as a link).

Reviewer 2 Report

In this data paper, the authors describe the carabid beetle species identified in Mordovia (Russia) since 1979. They summarize the occurrences obtained using different capture methods and compare them with existing bibliography to determine the relevance of the presence of different species.

Overall, this data paper is well written and does what it is asked to do: describe a dataset accurately. All the information needed to understand how the data was collected and stored in the dataset is present. With the exception of the capture method(s) per species to take into account the sampling effort. This would make it possible to compare species rarity with the associated sampling effort.

A good point is that the authors have taken care to add a small descriptive analysis of the species present and absent from the existing bibliography.

Minor comments:

One small improvement would be to modify the legends on Figure 1 to make them as clear as possible. "Occurrence by locality", "Journal of number of individuals" for example.

Please also add the north arrow

Author Response

Dear reviewer. We are grateful for the critical analysis of our manuscript and your recommendations. We tried to make all the corrections. We also made corrections in the figures.